# Personalized Prophylaxis with myPKFiT^CE^: A Real-World Cost-Effectiveness Analysis in Haemophilia A Patients

**DOI:** 10.3390/medicina60010034

**Published:** 2023-12-24

**Authors:** Ippazio Cosimo Antonazzo, Paolo Angelo Cortesi, Ezio Zanon, Samantha Pasca, Massimo Morfini, Cristina Santoro, Raimondo De Cristofaro, Giovanni Di Minno, Paolo Cozzolino, Lorenzo Giovanni Mantovani

**Affiliations:** 1Research Centre on Public Health (CESP), University of Milan-Bicocca, 20900 Monza, Italy; paolo.cortesi@unimib.it (P.A.C.); lorenzo.mantovani@unimib.it (L.G.M.); 2IRCCS Istituto Auxologico Italiano, 20145 Milano, Italy; 3Hemophilia Center, University Hospital of Padua, 35128 Padua, Italy; zanezio61@gmail.com; 4Laboratory Medicine, Department of Biomedical Sciences, Padua University Hospital, 35128 Padua, Italy; sampasca27@gmail.com; 5Italian Association of Haemophilia Centres—AICE, 50100 Firenze, Italy; massimo.morfini@unifi.it; 6Hematology, Umberto I University Hospital, 00161 Rome, Italy; santoro@bce.uniroma1.it; 7Center for Haemorrhagic and Thrombotic Diseases, Department of Medical Sciences, Catholic University School of Medicine, ‘A. Gemelli’ Hospital, 00168 Rome, Italy; raimondo.decristofaro@unicatt.it; 8Department of Clinical Medicine and Surgery, Regional Service Centre of Coagulation Disorders, ‘Federico II’ University, 80138 Naples, Italy; diminno@unina.it

**Keywords:** hemophilia A, health economics, pharmacokinetics-guided, prophylaxis

## Abstract

*Background and Objectives:* This study aimed to assess the effectiveness and costs associated with pharmacokinetics-driven (PK) prophylaxis based on the myPKFiT^®^ device in patients affected by hemophilia A (HA) in Italy. *Materials and Methods:* An observational retrospective study was conducted in three Italian hemophilia centers. All patients with moderate or severe HA, aged ≥ 18 years, capable of having PK estimated using the myPKFiT device, and who had had a clinical visit between 1 November 2019 and 31 March 2022 were included. Differences in clinical, treatment, health resources, and cost data were assessed comparing post-PK prophylaxis with pre-PK. The incremental cost-effectiveness ratio (ICER) was estimated as cost (EUR) per bleed avoided. *Results:* The study enrolled 13 patients with HA. The mean annual bleeding rate decreased by −1.45 (−63.80%, *p* = 0.0055) after the use of myPKFiT^®^. Overall, the consumption of FVIII IU increased by 1.73% during follow-up compared to the period prior the use of the myPKFiT. Prophylaxis based on the myPKFiT resulted in an ICER of EUR 5099.89 per bleed avoided. *Conclusions:* The results of our study support the idea that the use of PK data in clinical practice can be associated with an improvement in the management of patients, as well as clinical outcomes, with a reasonable increase in costs.

## 1. Introduction

Haemophilia A (HA) is an X-linked recessive, hereditary bleeding disorder resulting from deficiency or dysfunction of the coagulation protein factor VIII (FVIII), which is involved in coagulation processes [1]. It is a rare disease that affects one in 5000–7000 male births [1]. This coagulopathy manifests with bleeding episode in joints, soft tissues, and muscles [1,2]. HA can be classified into mild (FVIII level 5–40 international units per deciliter [IU/dL]), moderate (FVIII level 1–5 IU/dL), and severe (FVIII level < 1 IU/dL) based on circulating FVIII levels [1,2]. Patients with HA require lifelong therapy [1,2]. To date, the main treatments have been represented by replacement therapies, involving the administration of the deficient clotting factor to achieve hemostasis adequacy, and on-demand treatments, which are crucial for managing acute phase bleedings [1]. On the other hand, prophylactic treatment involves intravenous injections of factor concentrate to prevent bleeding and joint destruction, and aims at preserving the normal musculoskeletal functions [1]. Previous studies showed that prophylactic treatment, typically at a dosage of 20–40 international units/kilograms (IU/Kg) administered 2–3 times per week, was associated with reduced bleedings episodes, decreased hospitalizations, and improved long-term joint functions [2,3,4,5,6,7]. Primary prophylaxis usually starts at a very young age (≤2 years) before the onset of joint disease, whereas secondary prophylaxis starts after the onset of joint disease [8,9]. The main aim of prophylactic treatment is to increase the level of FVIII above 1 international unit per deciliter (IU/dL) concentration, which has been associated with lower risk of bleedings for patients [10,11]. However, achieving the target concentration in clinical practice is challenging and depends on individualization of treatment, as patients may experience bleeding episodes despite prophylaxis [12,13,14,15]. In this context, understanding the pharmacokinetics (PK) of FVIII replacement therapy is crucial to ensure its efficacy in treating these patients [16,17,18]. However, determining FVIII parameters requires several blood samplings (i.e., at minimum of 6 time points for children and 10 for adults) [19]. Given the challenges associated with blood sampling, PK-guide prophylaxis can difficult to routinely perform [20]. Bayesian models predicting PK parameters have shown promising results; accordingly, they have been suggested as a suitable alternative for PK estimates in a prophylactic context [21]. In particular, a medical device called myPKFiT^®^ was developed based on the aforementioned statistical method, to predict the PK parameters in patients treated with FVIII concentrate Advate^®^. This procedure requires fewer blood samplings (at least 2) and can be used by patients to calculate the appropriate dose of treatment to guarantee the optimal and efficient utilization of FVIIII concentrate during prophylaxis treatment [22,23,24].

Although myPKFiT^®^ was indicated as potentially benefiting patients, with no impacts on costs [25], few studies have assessed the impact of this device in prophylactic regimens in real-world settings. Therefore, this study aimed to assess the effectiveness and costs associated with the use of the myPKFiT^®^ device in patients with HA in Italy.

## 2. Materials and Methods

### 2.1. Study Design and Data Source

This was an observational retrospective study conducted in three Italian hemophilia centers: Azienda Ospedaliera—Ospedale di Padova Centro Emofilia, Clinica Medica II (Padova); Azienda Ospedaliera—Universitaria Policlinico Umberto I (Roma) U.O.C. Ematologia; and Azienda Ospedaliera Universitaria Policlinico Agostino Gemelli di Roma (Roma) U.O. di malattie emorragiche e trombotiche. Since this work was not intended as an interventional study, only patients for whom the utilization of myPKFiT^®^ was deemed suitable by their physician in routine clinical practice were included in the study. Therefore, as reported in a similar article [26], data inclusion in this study had no impact on the intended management, which was determined in accordance with the criteria of the treating specialist.

The study protocol and the informed consent form were approved by the Institutional Ethics Committees of all participating Centers. The study protocol adhered to the principles outlined by the 18th World Medical Assembly [27].

### 2.2. Inclusion and Exclusion Criteria

All patients with a diagnosis of moderate or severe HA (defined as plasma FVIII levels 1 to 5 IU/dL, and <1 IU/dL, respectively) [28], treated with a regimen of prophylaxis with Advate for at least 12 months, able to have PK estimated using the myPKFiT^®^ device, and who had had a clinical visit between 1 November 2019 and 31 March 2022 were selected. The date on which patients started to use the myPKFiT^®^ device was considered as the index date (ID). Patients for whom myPKFiT^®^ was not suitable, those with chronic liver disease, those with active cancer, and those unable to understand or sign the informed consent were excluded from the study.

For each included patient, data on clinical and demographic characteristics were gathered at ID and 12 months prior and after the ID.

### 2.3. Data Collection and Outcomes

For each included patient, data were collected retrospectively (12 months before ID and 12 months after ID). This encompassed demographic (i.e., sex and age) and clinical information, specifically the number of annual joint bleedings (AJBR), annual bleedings rate (bleeds/year) (ABR), the reason for the use of myPKFiT^®^ (i.e., reducing the treatment burden (reducing frequency or dose), evaluating reaching the target level, to maintain the FVIII target level and prevent bleedings, to increase the bleed protection due to change in life style (e.g., starting sport activities), other reasons), and orthopedic joint score—Hemophilia Joint Health Score (HJHS). Individual PK profiles were estimated using myPKFiT^®^. Specifically, myPKFiT^®^ blood samples were collected according to the device’s instructions (2.5–4.5 h and 22–23 h), and samples were assayed using one-stage or chromogenic FVIII:C assays.

Furthermore, the annual consumption of FVIII was recorded in the period prior and after the ID.

The annual consumption of FVIII was used to estimate the costs associated with the use of the myPKFiT^®^ device in the 1-year prior and after its implementation. We decided to estimate only the cost associated with FVIII consumption because, in this population, this represents up to 95% of the overall costs [29,30,31]. Hence, the annual costs related to FVIII consumption were computed by multiplying the mean of UI used by the price per IU of FVIII, as indicated by the Italian Medicine Agency (AIFA). The incremental cost-effectiveness ratio (ICER) between the period prior and after the use of the myPKFiT^®^ device was estimated by adopting the perspective of the Italian National Health Service (NHS). The ICER was expressed as cost in euros (EUR) per bleed avoided [29,30,31].

### 2.4. Statistical Analysis

Statistical estimates to describe the selected cohort in the two periods were shown as frequencies and percentages for categorical variables, mean and standard deviation (SD), or median and quartiles (Q_1_–Q_3_) for continuous variables. Difference in clinical outcomes was assessed by comparing after vs. before PK prophylaxis using the myPKFiT. Specifically, continuous data (i.e., ABR, AJBR) were compared within the two periods using a Wilcoxon sign rank test, whereas categorical variables (i.e., patients with zero bleeds) were compared using a McNemar test per repeated measures. A *p*-value < 0.05 was considered statistically significant. The resource utilization and associated costs within the two periods were reported as mean difference.

Finally, ICER was estimated as the variation in costs between pre- and post-use of myPKFiT^®^ device divided by the variation in the ABR and AJBR events within the same timeframe. Results were expressed as cost in euros per bleed avoided [29,30,31].

All analyses were performed using SAS software (version 9.4; SAS Institute Inc., Cary, NC, USA).

## 3. Results

The study enrolled a total of 13 male patients affected by HA. As shown in Table 1, the patient cohort was characterized by a median age (Q1–Q3) of 10.00 (9.00–38.00) years, a median weight of 50.00 (29.00–61.00) Kg, and a median height of 161.00 (130.00–169.00) cm. Patients were primarily underweight (46.15%) or had a normal weight (46.15%). Predominantly, the blood type observed was 0 Rh+ (39%), followed by A Rh+ (23%) and 0 Rh- (15%). Within the cohort, 7 patients were students (54%), 3 (23%) were employed full-time, and 3 were either part-time employed or unemployed.

Figure 1 shows the reasons for employing the myPKFiT^®^. A total of 39% of patients used the device to reduce bleeding, 31% to maintain the FVIII target level and prevent bleeding, and 15% to reduce the treatment burden, with the remaining individuals using it to increase bleed protection due to a change in lifestyle or other reasons.

As reported in Table 2, the number of bleeding events during the follow-up decreased by 16% compared to the year prior to the ID (data when patients started to use the myPKFiT^®^ device). In the year prior to the ID, only two patients had no bleeding events, whereas this number increased to seven patients during the follow-up. During the follow-up, the median ABR decreased from 2.00 (Q1–Q3: 1.00–4.00) in the year prior to the ID to 0.00 (0.00–1.00) during the follow-up (−100%) (*p*-value ≤0.05). In addition, the AJBR showed a significant reduction within the two study periods (*p*-value ≤ 0.05). A similar trend was observed in patients who experienced at least one bleeding event, but in this case the results were not statistically significant.

Table 3 shows changes in HJHS and in the treatment regimen. In the year after myPKFiT^®^ use, a 20% reduction (from 8.45 in the year prior to the ID to 6.82 during follow-up) in HJHS compared to the year prior to the ID was observed. The mean prophylactic dose increased up to 19% between the two study periods (from 36.2 prior to the ID to 43.0 during follow-up). Similarly, the UI consumed for prophylactic treatment by each patient increased by 5.6%. Conversely, the doses administered to manage bleeding events decreased by 70% between the study periods. Overall, the consumption of FVIII IU increased by 1.7% during the follow-up compared with the period prior to the ID.

Cost-effectiveness results are shown in Table 4. The treatment with myPKFiT^®^ yielded an ICER of EUR 5099.89 per bleed avoided. Specifically, the device resulted in a reduction in ABR events with an increased expenditure of EUR 7394.84 compared to the previous period (absence of the device). Similarly, the use of the device resulted in an ICER of EUR 7394.84 per joint bleed avoided (Table 4).

## 4. Discussion

Patients with HA face a higher risk of death, which is correlated with age and disease severity, compared to the general population [32]. For HA patients, prophylactic treatment with FVIII is considered a mainstay for reducing bleeding events, preventing joint damage, and improving quality of life [33,34,35,36]. However, the use of fixed-dose prophylaxis does not guarantee the eradication of bleedings [37,38]. Fixed-dose prophylaxis involves the administration of a predetermined amount of therapy at regular intervals to prevent bleeding episodes. Several factors might contribute to a lower efficacy of this approach. For example, individuals may have different bleeding phenotypes and responses to the treatment, and fixed doses prophylaxis might not account for clotting factor level variability. In addition, individuals might metabolize and eliminate the therapy at different rates, leading to variations in the duration of protection provided by a fixed dose. This might result, for some patients, in a suboptimal factor level, with a consequent increased risk of bleedings. Furthermore, patients with hemophilia may already have joint damage due to previous bleeding episodes. In this context, a fixed dose of prophylaxis might not be sufficient to address the challenge in maintaining joint health. Given the aforementioned challenges, personalized and adaptive approaches to prophylaxis, such as PK-guided dosing, might represent a valid alternative for treating patients with HA. Recently, several studies have documented the benefits associated with an individualized prophylaxis regimen guided by patients’ PK profiles [24,39,40,41]. Innovative devices have been developed to precisely determine optimal drug doses based on PK measurements, obviating the need for obligatory washout periods and minimizing sparse blood sampling.

Our findings show that PK-guided prophylaxis utilizing the myPKFiT^®^ device improved clinical outcomes. Specifically, the adoption of this device resulted in a reduction in bleeding episodes, ABR, and AJBR events in treated patients, all while limiting the overall increase in factor consumption. Although a small sample size is a common feature in hemophilia studies, further effects were not assessed in the current work because of the extremely limited number of patients, limiting the power of the statistical analyses. Nevertheless, our findings align with previous studies [24,42,43]. For example, in 2018, Mingot-Castellano and colleagues investigated the impact of the myPKFiT^®^ device in 36 patients with HA (13 aged ≤15 years and 23 aged >15 years) and found that use of the device was associated with a reduction in bleeding episodes, without an increase in factor consumption and resource optimization. In this context, the authors highlighted that the resources were optimized because the dose was reduced in patients who might have been previously over-treated and increased in patients who might have been previously undertreated [24]. Furthermore, the new device reduced the number of blood samples required to estimate the PK parameters, which represents a crucial factor for patients’ wellbeing, resulting in positive outcomes for their daily activities and their overall quality of life [24]. Additionally, Álvarez-Román et al. observed improved treatment adherence and the development of personalized prophylactic regimens in 27 patients utilizing the myPKFiT^®^ device [42]. A similar conclusion was drawn by Arvanitakis and colleagues (2021), who showed that, across different devices for factor prophylaxis, the use of the myPKFiT^®^ improved patient management, enabling the customization of treatments based on individual patient characteristics [43].

In our study, a cost-effectiveness analysis was conducted, a common approach for presenting results concerning the costs per clinical outcome for health technologies [44]. This methodology is essential for informing decision makers about the economic impact of a new technology, while considering its associated benefits. A cost-effectiveness analysis provides useful evidence for different purposes, such as future resource allocation. In this context, the limited resources generally available for healthcare systems force decision makers to allocate them for more effective treatments. Furthermore, a cost-effectiveness analysis can assist in identifying an intervention that might maximize health outcomes for a given investment. This aspect is crucial when multiple treatment options are available and decision makers need to choose the treatment most effective for a reasonable cost. In addition, cost-effectiveness analysis can be used by policy makers to design reimbursement policies, formulary decisions, and guidelines that promote the delivery of high-value care. In this study, we conducted a cost-effectiveness analysis to evaluate the clinical and economic impact of the myPKFiT device. Specifically, we assessed the cost associated with the use of the myPKFiT^®^ and both the number of bleedings and joint bleeds avoided through comparing the period before the device’s use with the period after. The findings indicated that the ICER for bleed avoided during prophylaxis treatment guided using myPKFiT^®^ data was EUR 5099.89. Similarly, the ICER for joint bleed avoided was EUR 7394.837. Our ICERs are in line with those observed by Gringeri and colleagues, who suggested an incremental cost of EUR 7537.00 per avoided bleed event during prophylaxis [29]. In addition, the PK-guided dosing prophylaxis resulted as cost-effective compared to standard prophylactic approaches in two other studies [25,45]. Specifically, Iannazzo and colleagues performed a cost-effectiveness analysis by comparing PK-driven prophylaxis with standard prophylaxis [25]. The authors found that the approach with the use of the myPKFiT device was preferable to a standard regimen because it was associated with better clinical outcomes (i.e., lower AJBR) and lower costs. In their study, the authors found that the use of the new device generated an overall cost saving of about EUR 5000 per patient-year and an ICER of about EUR −30,000 per bleed avoided [25]. Similarly, a cost-effectiveness analysis conducted in Gu and colleagues, in 2022, found that the costs associated with treatment injections were lower in PK-guided individualized prophylaxis compared with the standard one [45]. Similarly, the costs for the treatment of bleedings were also lower with the use of PK-guided prophylaxis compared with standard prophylaxis [45]. The authors found that the PK-guided prophylaxis was dominant (lower costs and more effective) compared with standard prophylaxis [45]. Although having some differences in study design, these findings corroborate the hypothesis that PK-guided individualized prophylaxis represents an opportunity to enhance the management of patients with HA, potentially leading to an improvement in their quality of life.

Our study contributes to the ongoing debate on the efficacy, safety, and costs associated with different treatment strategies for patients with HA, a topic of paramount importance for patients, physicians, and decision makers. The use of Bayesian models to predict PK parameters has shown promising results in this context. The myPKFiT device was developed to predict the PK parameters of FVIII in patients with hemophilia. It represents a support in clinician decision-making processes, considering the PK profile in combination with other individual factors such as bleeding phenotype, musculoskeletal system status, and bleeding risk associated with daily activities (i.e., physical activity) [24]. As other authors stated, the beneficial effect of personalized dosage of treatment can also improve a patient’s adherence to the treatment, with a consequent improvement in clinical outcomes. The findings of this study further support the potential benefits associated with the use of a PK-driven prophylaxis approach by adding data on its cost-effectiveness in real-world clinical practice, an area that remains poorly investigated. We believe that this body of evidence should stimulate scientific discussion and prompt a series of studies aimed at further exploring this topic in both European and non-European contexts, focusing on the differences between the standard and long-acting factors now available.

This study has some limitations. First, we included a small number of study subjects, but this is a well-known issue in studies on hemophilia, due to its rarity. Second, the retrospective nature of the study introduced the possibility of recall bias. However, it is important to note that this type of bias is more commonly associated with events having no or little impact on patients, whereas this study investigated severe and potentially fatal events, such as bleeding, registered at the center.

## 5. Conclusions

In the last decades, the treatment of hemophilia has significantly improved. In this context, the analysis of clinical outcomes and associated costs of new treatments is crucial for patients, clinicians, and decision makers. This study investigated the clinical and economic impact of the use of the myPKFiT in patients with HA. The use of the myPKFiT^®^ resulted in an improvement in different clinical outcomes, with a modest increasing overall FVIII consumption during follow-up compared with the year prior to the start of myPKFiT use. Notably, during the study period, the use of the device was associated with a reduction in ABR and AJBR. Furthermore, a prophylactic regimen guided by pharmacokinetic data resulted in a cost-effective treatment strategy compared with no PK approach. This corroborates the idea that the use of this data in clinical practice can contribute to an improvement in patient management and clinical outcomes, with a reasonable increase in costs. Furthermore, a PK approach could have a higher value when applied to the emerging extended half-life products. Future studies, conducted in other European and non-European settings, encompassing larger simple size and evaluating extended half-life products, should be performed to support the clinical and economic value of the PK-prophylaxis approach in routine clinical practice.

## Figures and Tables

**Figure 1 medicina-60-00034-f001:**
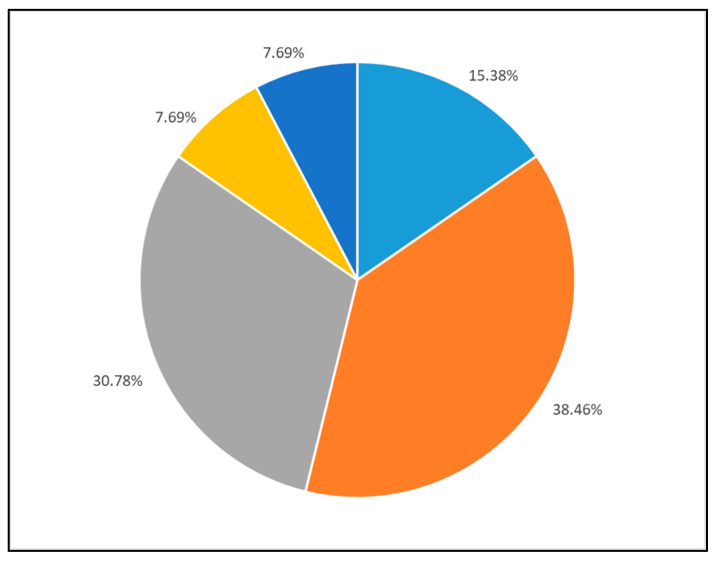
Reason for using myPKFiT device. Legend: light blue: reducing the treatment burden (educing frequency or dose); grey: to evaluate reaching target level; orange: maintaining FVIII target levels to prevent bleeding; yellow: increase the bleed protection due to change in life style (i.e., starting sport activities); dark blue: other reasons.

**Table 1 medicina-60-00034-t001:** Demographic and clinical characteristics of included patients with HA.

**N° of patients**	13
**Sex**, Male, N (%)	13 (100.00%)
**Age**, Median (Q_1_–Q_3_)	14.00 (9.00–38.00)
**Weight**, Kg—Median (Q_1_–Q_3_)	50.00 (29.00–61.00)
**Height**, cm—Median (Q_1_–Q_3_)	161.00 (130.00–169.00)
**BMI (Kg/m^2^)**	
Underweight (<18.50)	6 (46.15%)
Normal weight (18.50–29.99)	6 (46.15%)
Overweight/Obesity (≥25.00)	1 (7.69%)
**Blood Type**	
0 Rh−	2 (15.38%)
0 Rh+	5 (38.46%)
A Rh−	1 (7.69%)
A Rh+	3 (23.08%)
B Rh−	1 (7.69%)
B Rh+	1 (7.69%)
**Working status**	
Full-time employed	3 (23.08%)
Part-time employed	1 (7.69%)
Student	7 (53.85%)
Unemployed	2 (15.38%)

Q_1_–Q_3_: first quartile–third quartile; BMI: body mass index.

**Table 2 medicina-60-00034-t002:** Clinical outcome assessment in the observational period.

	One Year Pre-myPKFiT	One Year Post-myPKFiT	Δ (%)	*p*-Value
Total bleeds, N	25	9	−16	
Joint bleeds, N (%) *	15 (60.00)	4 (44.44)		
Patients with zero bleeds, N (%)	2 (18.18)	7 (63.64)	+5 (250.00)	0.0588
**All Patients**				
ABR				
Median (Q_1_–Q_3_)	2.00 (1.00–4.00)	0.00 (0.00–1.00)	−2.00 (−100.00)	0.0103
AJBR				
Median (Q_1_–Q_3_)	0.00 (0.00–3.00)	0.00 (0.00–1.00)	-	0.0271
**Patients with ≥1 bleed**				
ABR				
Median (Q_1_–Q_3_)	4.00 (4.00–6.00)	3.00 (1.00–4.00)	−1.00 (−25.00%)	0.1088
AJBR				
Median (Q_1_–Q_3_)	3.00 (3.00–4.00)	1.00 (1.00–2.00)	−2.00 (−66.67)	0.1088

* Percentage estimated on total bleeds (n. joint bleeds/total bleeds); Q_1_–Q_3_: first quartile–third quartile; ABR: annual bleedings rate AJBR: annual joint bleedings.

**Table 3 medicina-60-00034-t003:** FVIII consumption in the observational period.

	One Year Pre-myPKFiT	One Year Post-myPKFiT	Δ (%)
**HJHS Mean (±SD)**	8.45 (13.09)	6.82 (11.19)	−1.64 (−19.40)
**Prophylaxis**			
UI/Kg/Infusion Mean (±SD)	36.18 (13.83)	43.01 (13.46)	+6.83 (+18.88)
N Infusion per week, Mean (±SD)	2.91 (0.30)	2.55 (0.52)	
Annual prophylaxis consumption per patient, UI	210,941.70	222,792.40	+11,850.70 (+5.62)
**Treatment of bleeds**			
UI/Kg/Infusion Mean (±SD)	32.42 (9.43)	32.97 (5.16)	+0.55 (+1.69)
N of Infusion, Mean (±SD)	2.80 (1.41)	2.11 (1.17)	−0.69 (−24.60)
Annual bleeding consumption per patient, UI	11,450.00	3450.00	−8000.00 (−69.87)
**Total consumption per patient, IU**	222,391.70	226,242.40	+3850.70 (+1.73)

HJHS: hemophilia joint health score.

**Table 4 medicina-60-00034-t004:** Cost-effectiveness analysis results.

Treatments	Total Costs (EUR)	Δ Costs (EUR)	ABR	Δ ABR	ICER (Euros per Bleed Avoided)
**Pre-myPKFiT**	131,627.621	-	2.27	-	-
**Post-myPKFiT**	139,022.458	7394.837	0.82	−1.45	5099.89
**Treatments**	**Total costs (EUR)**	**Δ costs (EUR)**	**AJBR**	**Δ AJBR**	**ICER**
**Pre-myPKFiT**	131,627.621	-	1.36		
**Post-myPKFiT**	139,022.458	7394.837	0.36	−1.00	7394.837

ABR: annual bleedings rate, AJBR: annual joint bleedings.

## Data Availability

The data can be shared upon reasonable request.

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
