# Peer review of "Personalized Prophylaxis with myPKFiTCE: A Real-World Cost-Effectiveness Analysis in Haemophilia A Patients"

_medicina, 2023, doi:10.3390/medicina60010034_

Round 1
Reviewer 1 Report (Previous Reviewer 1)
Comments and Suggestions for Authors
Line 94-95: Definition of moderate and severe hemophilia : it is usually factor levels 1-5% (0.01-0.05 IU/ml) and <1 % (<0.01IU/ml) is the standard. Is there reference for 1-2 and >1 IU/dl to be used as cut off? If so please provide that.
Minor grammatic error:
Line 140: according to instruction (not by)
Comments on the Quality of English LanguageGood.
Author Response
We thank the reviewer for the comments on our manuscript which gave us the possibility to improve it. In the new version of the manuscript we carefully considered all raised points.
Our point-by-point responses are provided below.
Comments and Suggestions for Authors
Line 94-95: Definition of moderate and severe hemophilia : it is usually factor levels 1-5% (0.01-0.05 IU/ml) and <1 % (<0.01IU/ml) is the standard. Is there reference for 1-2 and >1 IU/dl to be used as cut off? If so please provide that.
We thank the reviewer for the crucial comment that provided us with the opportunity for a double check. We agree with the reviewer’s comment. The cutoffs used in the study were those also recommended by the reviewer. Unfortunately, there was a mistake when we reported the parameters in the paper. In the new version of the paper, we have updated the sentence to align with the parameters used in the study.
Minor grammatic error:
Line 140: according to instruction (not by)
Amended.
Reviewer 2 Report (Previous Reviewer 2)
Comments and Suggestions for Authors
Authors have been responsive to reviewer comments
Author Response
Authors have been responsive to reviewer comments
We thank the reviewer for the comment.
Reviewer 3 Report (Previous Reviewer 3)
Comments and Suggestions for Authors
The original version of the manuscript was revised by the authors. They have addressed and corrected all the issues I previously mentioned.
It it must be emphasized that the authors have introduced a new paragraph in which they clearly mention the usefulness of this study.
Minor requirements:
Lines 94-95 Moderate hemophilia: 1% to 5% factor level. Severe hemophilia: < 1% factor level. Please revise your statement. (https://www1.wfh.org/publications/files/pdf-1863.pdf )
Comments on the Quality of English Language
Line 100: those unable to understand or sign the informed consent
Line 103: (myPKFiT® device start use) it is not necessary. The significance of ID is clearly mentioned five lines above.
Line 134: „number of annual bleedings rate” should be „annual bleedings rate”
Line 136-142: please check the grammar.
Line 170: the 39% correspond to 0 Rh+. Please revise because 0 Rh- is written twice.
Table 2 and Table 4 (legend) It is written AJRB instead of AJBR.
Thank you!
Author Response
We thank the reviewer for the comments on our manuscript which gave us the possibility to improve it.
Our point-by-point responses are provided below.
Comments on the Quality of English Language
Line 100: those unable to understand or sign the informed consent
We thank the reviewer. In the new version of the manuscript we have correct the issue.
Line 103: (myPKFiT® device start use) it is not necessary. The significance of ID is clearly mentioned five lines above.
Amended.
Line 134: „number of annual bleedings rate” should be „annual bleedings rate”
Amended.
Line 136-142: please check the grammar.
Amended. In addition, we have also checked the entire manuscript.
Line 170: the 39% correspond to 0 Rh+. Please revise because 0 Rh- is written twice.
Amended.
Table 2 and Table 4 (legend) It is written AJRB instead of AJBR.
Amended.

This manuscript is a resubmission of an earlier submission. The following is a list of the peer review reports and author responses from that submission.
Round 1
Reviewer 1 Report
Comments and Suggestions for Authors
Inclusion:
Line: 85: Define moderate and severe hemophilia
Also inclusion criteria mentions 18 yrs or older but median age in table 1 is 14 (ranging from 9-38yrs) – Can you please confirm age eligibility for the study? As majority bleeding events begin to occur in childhood if age was chosen as 18 can you explain the rationale?
Please clarify the time point at which patient eligible for study: At diagnosis, minimum no of exposure days to FVIII etc..,
What assay was used for Factor VIII measurement?
Also please clarify timelines used for PK assessment( baseline, 4hr after infusion till what timeline 48-72h post infusion etc..,) and when intervention was done by treating physician- Based on Peak/trough level, AUC etc..
Table:1 Baseline characteristics: Please include individual patient characteristics If possible as it is a small 13 patient study population. In addition to demographic characteristics please also include disease characteristics: How many were severe and moderate hemophilia A, Average Exposure days to Factor VIII prophylaxis, dosing interval for prophylaxis, BMI etc..,
Discussion can be enhanced with why your study is still relevant when there is already published literature with use of PKFIT in prophylaxis setting- like including cost analysis which has limited data in literature.
Comments on the Quality of English LanguageSeveral grammatical errors and typos throughout the manuscript.
eg: Inclusion criteria: Line 89: Not able to understand or signed consent ( should be unable to sign consent)
Conclusion: Line 212: Improved different clinical outcome: Would be more meaningful if the outcomes measured reported(ABR, AJBR )
Reviewer 2 Report
Comments and Suggestions for Authors
This is an interesting and informative manuscript.
1. The manuscript would be improved by stating units such as bleeds/year for annual bleeding rate.
2. The authors should consider either providing a table of abbreviations or being careful to state the meaning of abbreviations the first time they are used.
3. For readers outside of Europe, it would be helpful to state the study start/end dates as November 1, 2019/March 31, 2022 to avoid confusion about the date abbreviation conventions.
4. I do not doubt that the modest cost per bleed prevented is cost effective. However, the authors should provide a citation about standard definitions of cost effectiveness if they can identify such a citation. If they cannot, it might be helpful to state the typical hospitalization or other resource cost associated with a "typical" hemophilia bleed.
Reviewer 3 Report
Comments and Suggestions for Authors
The manuscript is an original study. The main objective was to asses the cost-effectiveness of using PK-guided prophylaxis in patients with moderate and severe HA.
The results are important because bleeding is the main contributor to morbidity in these patients. In the long-term these are reflected in the reduction of the quality of life, the increased need for medical treatments and the reduction of the ability to work.
The manuscript has many strong points:
- The enrolled patients are young, therefore they have a long life expectancy and it is assumed that they lead an active life (with risk of joint bleeding). They are the population that would benefit the most from adequate prophylaxis.
- The study protocol is clearly presented.
- The discussion section frames current findings into available knowledge and underlines the importance and practical relevance of the study.
Some minor issues need to be addressed:
Row 128: blood type was 0 Rh- (39%), followed by a Rh+ (23%) and 0 Rh- (15%).
According to Table 1 it is: blood type was 0 Rh+ (39%), followed by A Rh+ (23%) and 0 Rh- (15%).
Figure 1 needs revision. There is not an adequate correspondence between percentages and size of slices.
Thank you!
Comments on the Quality of English LanguageRow 37: please consider: bleeding in joints, soft tissues, and muscles
Row 88: Patients under treatment with FVIII inhibitors (Please revise. In this form it results that the patients receive treatment with FVIII inhibitors)
Row 132-134. Please rewrite this sentence (the part: 31% reducing bleeding achieving and maintaining FVIII target levels to enhance bleed protection). The same issue in the Legend of figure 1.
Thank you!